# Rationing of Nursing Care in Intensive Care Units

**DOI:** 10.3390/ijerph17196944

**Published:** 2020-09-23

**Authors:** Agnieszka Młynarska, Anna Krawuczka, Ewelina Kolarczyk, Izabella Uchmanowicz

**Affiliations:** 1Department of Gerontology and Geriatric Nursing, Faculty of Health Sciences in Katowice, Medical University of Silesia,40−752 Katowice, Poland; mlynarska83@gmail.com (A.M.); krawucz@poczta.onet.pl (A.K.); 2Department of Propaedeutics of Nursing, Faculty of Health Sciences in Katowice, Medical University in Silesia, 40−752 Katowice, Poland; 3Division of Nursing in Internal Medicine Procedures, Department of Clinical Nursing, Faculty of Health Sciences, Wroclaw Medical University, 51-618 Wroclaw, Poland; izabella.uchmanowicz@umed.wroc.pl

**Keywords:** missed nursing care, rationing nursing care, anaesthesiological nurse, intensive care unit, PRINCA (Perceived Implicit Rationing of Nursing Care), MFIS (Modified Fatigue Impact Scale)

## Abstract

The nursing practice refers to a wide range of tasks and responsibilities. In a situation where there is a problem of limited resources, nurses are forced to ration the patient’s care—that is, minimize and skip some tasks. The main purpose of this work was to assess the rationing level of nursing care among staff in the intensive care units. Methods: The research included 150 anaesthesiological nurses in the Silesian Region in Poland. The research was conducted from July to October 2019 using the standardized Perceived Implicit Rationing of Nursing Care (PRINCA) questionnaire on rationing nursing care, assessing the quality of patient care, and job satisfaction. The Modified Fatigue Impact Scale (MFIS) standardized questionnaire was used to assess the level of fatigue of respondents in the physical, cognitive, and psychosocial spheres. Results: Sociodemographic factors, such as gender, age, place of residence, education, seniority, and type of employment were not found to affect the rationing level of nursing care in the intensive care unit. The average quality of patient care was 6.05/10 points, while the average job satisfaction rating was 7.13/10 points. Analysis of the MFIS questionnaire showed that respondents experienced fatigue between “rare” and “sometimes”, and nursing staff fatigue was the main factor for rationing care. Conclusions: The higher the level of fatigue, the greater the rationing of care and the less satisfaction from work.

## 1. Introduction

The health care services provided as part of anesthesiology and intensive care in healthcare entities include anesthesia, intensive care, resuscitation, and pain management [1]. In intensive care units (ICUs), patients who are critically ill, or on the verge of life and death, are treated. The main role of nurses working at ICUs is 24-h monitoring of patients’ vital functions, nutritional treatment to prevent the development of malnutrition in unconscious patients [2,3], ensuring proper personal hygiene, and keeping records of hospitalized patients [2,4].

A description of the health care services provided under anaesthesiology and the intensive care in healthcare entities includes, in accordance with the anesthesia and the intensive care standards of the Polish Society of Anaesthesiology and Intensive Care, that in “departments with the highest level of care, the ratio of the number of nurses per shift to the number of patients should be 1:1, that is, the approximate number of nurses per one hospital bed is six. In the case of a lower level of care, the number of nurses in relation to the number of patients should be 1:1.5 (2:3), or the number of nurses needed for one bed is four” [4]. A lack of nurse staffing leads to rationing of nursing care, which leads to poorer patient outcomes [5].

Rationing nursing care refers to the suspension or discontinuation of certain aspects of care due to limited resources, including time, staff shortages, and incorrect allocation of tasks [5,6]. 

In general, it can be assumed that adverse working conditions lead to rationing of nursing care interventions. In turn, limiting or skipping interventions in individual patients may affect patient results [7,8,9,10].

The issue of rationing nursing care was studied by Schubert et al. (2008), who stated that the risk of patient quality and safety during treatment or nursing care significantly reduces the level of patient satisfaction [11]. Many nurses point to the problem of overwork and too much workload in the process of providing nursing care. The test results say that nurses do not have enough time to perform the necessary nursing tasks [12]. Therefore, rationing of care causes adverse events, such as some of the most common events, including treatment errors, falls, infections, pressure sores, and critical incidents [9,10]. The relationship between adverse events in nursing care rationing and fatigue in intensive care units is not yet known.

There are a lot of studies which explored the problem of rationing nursing care in the general population of nurses; however, there has been a lack of investigation on the specific nursing work environment, such as anaesthesiologiclal nursing care. In neonatal intensive care, the reported outcomes of rationing nursing care in nursing interventions appear to influence parent and infant readiness for discharge, as well as pain control [13]. Fatigue refers to an overwhelming sense of tiredness and lack of energy, as well as a feeling of exhaustion associated with impaired physical and/or cognitive functioning [14]. Fatigue in nurses’ work may implicate consequences like avoiding contact with patients, a negative self-assessment of performance, and a host of other responses that may adversely impact on personal and professional well-being and functioning [15]. According to the International Affairs and Best Practice recommendations, researchers should conduct studies regarding the relationship between fatigue, workload, work hours, and the amount of sleep needed to provide safe patient care. It is recommended to explore the relationship between nurse fatigue and patient safety, identify measures to decrease fatigue, and reduce the impact of fatigue on both patient and nurse safety [16]. This study aimed to assess the rationing level of nursing care among staff in the intensive care unit. Despite the theoretical rationale and the growing interest in the concept of rationing nursing care with an increasing number of studies demonstrating a significant relationship between rationing nursing care and patient outcomes or between the nursing practice environment and patient outcomes, significant gaps remain [17,18].

An important gap is the identification of nursing practice environment factors that are associated with missed nursing care in the intensive care units. To address these gaps in our knowledge, we sought to explore the relationship between specific factors of the anaesthesiological nursing practice environment and missed nursing care. Therefore, the main purpose of the study was to assess the rationing of nursing care in anaesthesiological nursing practice and related factors of missed nursing care in Intensive Care Units. 

## 2. Materials and Methods 

### 2.1. Study Design

In our study, a cross-sectional survey was conducted. The sample included 150 anaesthesiological nurses working at the intensive care units located in the Silesian Region in Poland. The data for the present study were collected from July to October 2019. 

### 2.2. Instruments

Data were collected using an author’s survey questionnaire consisting of 13 questions constituting the metric, and three open and 10 closed questions. To measure the main variables, the Perceived Implicit Rationing of Nursing Care (PIRNCA) [19] and Modified Fatigue Impact Scale (MFIS) [20] surveys were used.

The PIRNCA standard questionnaire consisted of 31 questions regarding rationing of nursing care and two questions about the nursing assessment of the quality of patient care and job satisfaction. In each of the 31 questions in this part of the questionnaire, the following point scale was adopted (according to the key to it): “never”(0), “rarely”(1), “sometimes”(2), and “often“(3). The final result is the average of points from questions in which one of the above answers is marked (i.e., questions of which “not applicable” is excluded). Thus, the total result is a number in the range of 0–3, and can also be interpreted according to the above scheme. In the two questions about the nursing assessment of the quality of patient care and job satisfaction, the scale of answers was from 0 to 10, where higher numbers meant better nursing assessment of the quality of patient care and greater job satisfaction. The translated PIRNCA questionnaire indicates a high level of reliability and validity, fully comparable to that of the original. The questionnaire can be used for the assessment of PIRNCA in Polish hospitals [19].

The standardized MFIS questionnaire allows to assess the overall level of fatigue of respondents and their fatigue in the physical, cognitive, and psycho-social areas. Each subscale of the MFIS questionnaire has a different number of questions. Hence, each scale has a different range of values. On each of them, however, a higher number of points indicate high fatigue. There are no standards to tell how many points correspond to high fatigue. However, for each subscale, it is possible to calculate the average number of points per question and interpret according to the key to a single question, in which (0) means “never”, (1) “rare”, (2) “sometimes”, (3) “often”, and (4) “almost always”. It is considered to be a good tool for assessing the impact of fatigue on people’s lives, and has been shown to have good internal consistency, validity, and reproducibility [21].

### 2.3. Participants

The criterion for inclusion in the study was full employment in the intensive care unit over six months, and the profession of a nurse. The criterion for not being qualified for the study was the lack of consent to participate in the study, and an incomplete survey.

### 2.4. Data Collection Procedures and Statistic Procedure

Data were collected in a database in the form of a Microsoft Excel spreadsheet. The results were analyzed using the R Core Team program for statistical analysis (version 18.1, 2019, R Foundation for Statistical Computing, Vienna, Austria). The analysis of quantitative variables (i.e., expressed in numbers) was performed by calculating the mean, standard deviation, median, quartiles, and minimum and maximum. The analysis of qualitative variables (i.e., not expressed in numbers) was performed by calculating the number and percentage of occurrences of each value. The comparison of quantitative variable values in two groups was performed using the Mann-Whitney test. Comparison in three or more groups was made using the Kruskal-Wallis test. Correlations between quantitative variables were analyzed using the Spearman correlation coefficient. Multifactorial analysis of the independent impact of many variables on the quantitative variable was performed by linear regression. The results are presented as the values of the parameters of the regression model with a 95% confidence interval. A *p*-value of <0.05 was considered statistically significant.

### 2.5. The Ethical Procedure

It was explained that nurses’ participation in the research was voluntary and anonymous. Their permission and written consent were requested and obtained before assessment. Approval from the Bioethical Commission of the Silesian Medical University in Katowice was obtained prior to the commencement of this study (Ethical Number: PCN/0022/KB/17/20).

## 3. Results

The average age of the study group was 38.75 ± 9.1 years, and the majority were females (139/150 or 92.67%). In the question about rationing care, the average number of points obtained was 0.81 ± 0.68, which means that it is “rarely” rationed. The average quality of patient care was 6.05 ± 1.67 points. Therefore, the average job satisfaction rating was −7.13 ± 1.72 points. There was no statistically significant difference between rationing of care and demographic factors of the nurses examined. The rationing of nursing care and demographic characteristics of participants are reported in Table 1.

### 3.1. The Characteristics Features of the Nurses’ Health

Diseases that the respondents observe at home mainly included back pain syndromes (95/150 or 63.33%), fatigue (94/150 or 62.67%), and headaches (65/150 or 43.33%). Respondents also indicated sleep disorders (56/150 or 37.33%), varicose veins of the lower extremities (41/150 or 27.33%), hair loss (43/150 or 28.67%), and skin problems (46/150 or 30.67%). Digestive problems were reported by 31/150 (or 20.67%) of respondents, and burnout was felt by (26/150 or 17.33%). Additionally, 21/150 or 14% were obese, and 20/150 or 13.33% of respondents admitted that they suffered from thyroid disease, and 17/150 or 11.33% from cardiovascular disease. The vast majority (113/150 or 73%) had not been on sick leave during the last year due to illness, and one person did not respond. Among the substances used by nurses, 122/150 (or 81.33%) drank coffee, 32/150 (or 21.33%) smoked cigarettes, and energy drinks were consumed by 10/150 people (or 6.67%).

### 3.2. The Characteristics of Working Conditions

Over half of the respondents (39/150 or 54.67%) were, on average, satisfied with their financial situation. Regarding patients per nurse, half of the respondents (76/150 or 50.67%) replied that they had two patients. However, as many as 46/150 (or 30.67%) of respondents took care of over two patients. The vast majority of respondents (115/150 or 76.67%) believed that the number of nurses employed in the ward was not sufficient to provide comprehensive care for the sick. As many as 116/150 respondents (77.33%) were not able to regularly eat meals at work. In the case of drinking water during the on-call time, only 5/150 people (or 3.33%) admitted that they drank more than 2 litres; the rest of the participants said they drank less than 1 litre (77/150 or 51.33%). According to the vast majority of respondents (123/150 or 82%), the lifting standards in the workplace were not respected. Only 31/150 (or 20.67%) used equipment to facilitate the lifting of patients. Almost half of the respondents (73/150 or 48.67%) had proper knowledge of lifting norms for women and men. The most common harmful factors indicated by nurses to which they were exposed during their duties were: detergents and disinfectants (143/150 or 96.33%), direct contact with chronically ill people (132/150 or 88%), ionizing rays, lasers (116/150 or 77.33%), contact with devices connected to high voltages (105/150 or 70%), and microorganisms (102/150 or 68%). The lowest number of respondents—only five people (or 3.33%)—indicated psycho-mental burdens. If it were possible to change work to another department, only 28 respondents would decide to do so (or 28%).

### 3.3. The Nursing Practice Environment and Rationing of Nursing Care

The average number of respondents in the nursing profession was 15.73 ± 9.94 years, and ranged from 11 months to 36 years. Seniority in the intensive care unit was, on average, 11.38 ± 9.01 years, and ranged from 0 to 31 years. The number of jobs held was also analyzed. Most respondents confirmed that they worked in an additional workplace; others (65/150 or 43.33%) had no additional employment. The highest percentage of respondents (127/150 or 84.67%) admitted to working 12-hour shifts. The remaining respondents were one-shift employees (15/150 or 10%), and eight respondents (5.33%) worked in an 8-hour system. The results obtained are presented in Table 2. No statistically significant differences were found between the rationing of care and the professional characteristics of the nurses surveyed.

### 3.4. MFIS and Rationing of Nursing Care

In the general fatigue questions, the average MFIS total score is 30.9 points, which gives 1.47 points per question, so the respondents’ experiences of fatigue were between “rare” and “sometimes”. However, the average score on the scale of physical functions is 13.4 points, which gives 1.49 points per question; therefore, respondents experience fatigue in this range between “rare” and “sometimes”. The average score on the cognitive scale is 13.99 points, which gives 1.4 points per question; consequently, respondents experience fatigue in this range between “rare” and “sometimes”. The average score on the scale of the psycho-social functions is 3.5 points, which gives 1.75 points per question, thus respondents experience fatigue in this range between “rare” and “sometimes”. The combined MFIS score correlates significantly (*p* ˂ 0.05) and positively (r ˃ 0) with care rationing, and therefore the more fatigue, the more frequent rationing of care A1. The total MFIS result correlates significantly (*p* ˂ 0.05) and negatively (r ˂ 0) with the assessment of job satisfaction, as a consequence the bigger the fatigue is, the lower the job satisfaction. The result on the subscale of psycho-social functions correlates significantly (*p* ˂ 0.05) and positively (r ˃ 0) with care rationing, that is, the greater the psycho-social fatigue, the more frequent the care rationing. The details of the results obtained are presented in Table 3.

According to the linear regression model, the MFIS total score, where the regression parameter was 0.021, is a statistically significant independent predictor of rationing nursing care. The details of the data obtained are presented in Table 4. In the linear regression model, none of the analyzed features is an important independent predictor of the quality of patient care (Table 5) and job satisfaction (Table 6).

## 4. Discussions

The analyses of this research are important in the context of the quality of nursing services. Knowledge of the reasons for rationing nursing care allows one to implement appropriate measures and prevent the negative effects of rationing care. This includes both the patients’ health and the professional career of nurses [22].

Research conducted for the purposes of this study showed that nurses rationalized medical care due to fatigue. A tired nurse has a tendency to ration her care and is less satisfied with her work. Research has shown that variables that do not affect rationing are issues related to age, seniority, or the need for shift work. In a study of nurses in Texas, 38% of the respondents reported that they committed a fatigue-related error that could have impacted patient safety. Nurses should not work while fatigued, because it can damage their ability to provide safe, competent, empathetic, and conscientious care to the patients [23]. As reported in the literature, fatigue can be exacerbated by increased numbers of shifts worked without a day off. In addition, working more than four consecutive 12-hour shifts is associated with excessive fatigue and longer recovery times. [24] According to our study, 84.67% respondents worked 12-hour shifts. Further, all studied fields of fatigue (physical, cognitive, and psycho-social) were significantly correlated with the rationing of nursing care (*p* < 0.005).

Participants aged 34 years and younger reported more MNC than those in the age groups of 45–54 years (*p* < 0.01) and those aged 55 years and older (*p* < 0.01). RNs reported significantly more MNC than did PNs (t(525) = 5.046; *p* < 0.001).

The predictors of rationing nursing care in Icelandic hospitals were studied by Bragad’ottir et al., using a combined questionnaire of the MISSCARE Survey-Icelandic and the Nursing Teamwork Survey-Icelandic (NTS-Icelandic). They found that missed nursing care was significantly lower in intensive care units than in medical and surgical units. The rationing of nurses’ care was found in the age group of 34 years and younger who reported more missed nursing care than those in the age groups of 45–54 years and older [25]. In our study, almost half of participants were between 20–40 years old, and we did not find any significant difference between the age, the rationing care, the quality of patients’ care or job satisfaction.

In these studies, such factors as education, continuing professional development education, number of jobs held, and the work system did not significantly influence the rationing of nursing care (*p* > 0.005). Slightly different conclusions were drawn from studies that were organized by Schubert et al., where 35 Swiss hospitals that had intensive care units in their structure participated in this study. Of the nurses surveyed, 98% said that during the last seven business days, at least one of the 32 nursing tasks listed in BERNCA (a tool to assess the rationing level of nursing care) had to be rationed. The average level of rationing was 1.69 ± 0.571, and which indicated that nurses “rarely” reported the inability to perform nursing tasks listed in BERNCA [26]. Multilevel regression analysis confirmed that the factors that affect rationing of care were issues related to staffing change and a sense of security in a given hospital facility. The smaller number of nurses in the ward meant that the nurses were not able to perform all the tasks specified in BERNCA. In our research, we did not check the number of shift staff in the intensive care unit, while it is a characteristic unit that differs from other departments. We also did not examine the nurses’ sense of security. We focused on other nurse factors important for the care of an anesthesia patient. Contrary to popular assumptions, factors related to the burden of professional duties, the nurse’s experience, and the nurse’s education did not affect the regulation of healthcare provided to patients. The results of Schubert et al. partly overlap with the data obtained in our own research. In our study, the largest group was nurses with work experience up to 5 years, with an average age of 38.75 ± 9.1 years, and these factors did not significantly affect the rationing of nursing care. Swiss research also shows that age and seniority are not predictors of rationing medical care [26].

The present study presents the predictors of nursing care rationing characterized in the PRINCA questionnaire, and shows that the greatest factor in nursing care rationing in intensive care units is fatigue. The Swiss results proved that the frequency of rationing differed between 32 BERNCA positions, which indicates different priorities of necessary nursing tasks. The identified predictors of rationing, adequacy of staff resources, and safety climate indicate actions that should be taken to improve the situation in the local healthcare system—namely, researchers recommend proactive changes to improve the adequacy of staff resources to reduce the risk of negative impact on patient results [26]. Such reflections are universal and can also be used in Polish hospitals, but most of all, the problem of nurse fatigue should be taken into account as the main factor in rationing care in Polish intensive care units.

The PRINCA questionnaire gives us the opportunity to answer how much the level of satisfaction with the work performed affects the rationing of nursing care. In our study, the rationing of nursing care in the Intensive Care Unit was associated with low job satisfaction, which was statistically significantly correlated with fatigue on the MFIS scale. Our research can be referred to the study of Papastavrou et al. The results of these authors reveal that the workload of nurses, patients’ behavior, and related communication barriers can be potential causes of rationing. In the detailed part, which analyzed individual elements of the work of nursing staff, it was shown that the reason for rationing care was mainly low job satisfaction and a sense of professional failure. In addition, rationing was associated with the organization of work in the ward in which nurses provided their services [27]. The cited results, similarly to the results of our research, indicate that rationing of nursing care is determined by low satisfaction with the performed work and burden of duties.

The mentioned studies by Schubert et al. [26] and Papastavrou et al. [27], which were related to nursing work in various departments, are consistent with the results of our work, where age and seniority do not significantly affect the rationing of nursing care. The research focused on the rationale for nursing care in neonatal Intensive Care Units. Rochefort et al. pointed out that staff shortages and non-supportive work environments are the reason for rationing nursing interventions. The results obtained during this analysis were interesting from the perspective of the research conducted in this work, and allowed to partially confirm them. It has been proved that rationing medical care is conducive to children’s general condition. It has been shown that the closer the patient is from being discharged from the hospital, the greater the tendency to ration nursing care [13]. Similar to our work, it has not been shown in these studies that rationing of medical care is influenced by such variables as the age of the nurse or professional experience.

Bachnick et al. contributed to the research on Intensive Care Units, and pointed out in their research appropriate working conditions in the context of the rationing of nursing care, such as staffing, staff management, and the level of implicit rationing of nursing care. Studies of these authors have shown, among others, that a higher level of alleged rationing of nursing care was associated with lower levels of patient-centred care, such as tailored treatment and care (β −0.912, 95% CI: −1.50–0.33). According to the authors, in order to reduce the level of hidden rationing of nursing care, it is necessary to improve working conditions and relieve nursing staff [28]. In our research, we did not focus on staff management; our goal was to cover other related factors of rationing nursing care in Intensive Care Units. We also did not focus on personal competences of nurses related to job satisfaction and rationing of nursing care, like Jaworski et al. The researchers showed that optimistic thinking, and job and life satisfaction significantly affected the level of implicit rationing of nursing care. Nurses from the “pessimistic” group were statistically significantly more likely to ration nursing care than nurses from the “optimistic” group. According to the authors of this research project, activities that consist in strengthening personal competences, and supporting and responding to all identified needs can contribute to increasing the satisfaction of nurses with work, and thus reduce the risk of rationing nursing care [29]. In our study, the main factors which correlated with job satisfaction were physical and cognitive fatigue, and general fatigue.

### 4.1. The Implication for Nursing Practice

The cited results of studies from Poland and other countries show that the reasons for rationing nursing care are usually associated with the way nurses are organized and their job satisfaction. For this reason, it is necessary to control the phenomenon of rationing nursing care and implement solutions that will relieve healthcare professionals, and thus increase their satisfaction with their work. These studies are significant in clinical practice, as they reveal new factors like in rationing nursing care in the Intensive Care Unit. The level of fatigue related to job satisfaction should be considered in implementing a prevention strategy of rationing nursing care.

### 4.2. The Limitation

Study limitations include the small sample size of the analyzed group of anaesthesiological nurses (n = 150). This research is the first step in expanding the knowledge in this area; thus, in the future, we plan to collect more study groups in a multi-centre study. Replicating this study with more vast data and more robust study designs is warranted to confirm these results. These studies also need to be extended to include such factors as management staff, and staffing and working conditions as predictors of rationing of nursing care in the Intensive Care Unit.

## 5. Conclusions

The sociodemographic factors, education and seniority of nurses, type of employment, and the number of jobs do not affect the rationing of care. Undoubtedly, staff fatigue affects the rationing of nursing care. The higher the level of fatigue, the greater the rationing of care, and the less satisfaction from work.

## Figures and Tables

**Table 1 ijerph-17-06944-t001:** Demographic characteristics of nurses compared with the rationing of nursing care.

Characteristics Feature	Participations (*n* = 150)	Princa		
		Rationing Care	Quality of Patient Care	Job Satisfaction
Sex, *p*-value ^1^		*p* 0.739	*p* 0.585	*p* 0.895
Females, *n* (%), M ± SD	139 (92.67%)	0.81 ± 0.68	6.05 ± 1.68	7.12 ± 1.74
Males, *n* (%), M ± SD	11 (7.33%)	0.7 ± 0.65	6 ± 1.61	7.27 ± 1.56
Age, *p*-value ^2^		*p* 0.132	*p* 0.979	*p* 0.822
20–30 years old, *n* (%)	31 (20.67%)	r = −0.131	r = 0.002	r = 0.019
31–40 years old, *n* (%)	47 (31.33%)			
41–50 years old, *n* (%)	63 (42.00%)			
51–60 years old, *n* (%)	9 (6.00%)			
Marital status, *p*-value ^1^		*p* 0.864	*p* 0.923	*p* 0.793
Single, *n* (%), M ± SD	41 (27.33%)	0.81 ± 0.67	6.02 ± 1.68	7.22 ± 1.55
Married, *n* (%), M ± SD	99 (66.00%)	0.8 ± 0.68	6.06 ± 1.67	7.09 ± 1.81
Divorcee, *n* (%)	6 (4.00%)			
Widow, *n* (%)	4 (2.67%)			
Education, *p*-value ^3^		*p* 0.263	*p* 0.619	*p* 0.818
Medium, *n* (%), M ± SD	17 (11.33%)	0.72 ± 0.79	5.7 ± 1.44	7.24 ± 2.08
Bachelor’s degree, *n* (%), M ± SD	57 (38.00%)	0.9 ± 0.66	6 ± 1.71	7.05 ± 1.8
Master’s degree, *n* (%), M ± SD	75 (50.00%)	0.75 ± 0.67	6.14 ± 1.69	7.17 ± 1.6
Doctoral, *n* (%)	1 (0.67%)			
Continuing professional development education, ^1^				
Qualification course of nursing, anesthesia and intensive care, *n* (%), *p*-value ^4^	94 (62.67%)	0.73 ±0.65 *p* 0.136	5.95 ± 1.71 *p* 0.373	7.01 ± 1.76 *p* 0.321
Specialist course, *n* (%), *p*-value ^4^	60 (40.00%)	0.8 ± 0.66 *p* 0.987	6.12 ± 1.66 *p* 0.578	7.13 ± 1.81 *p* 0.959
Specialization of anaesthesiological nursing and intensive care, *n* (%), *p-value* ^4^	77 (51.33%)	0.83 ± 0.74 *p* 0.995	6.13 ± 1.63 *p* 0.413	7.26 ± 1.71 *p* 0.385
Postgraduate education, *n* (%), *p*-value ^4^	14 (9.33%)	0.63 ± 0.57 *p* 0.489	6.64 ± 1.82 *p* 0.136	7.21 ± 1.81 *p* 0.856
Non-applicable, *n* (%)	15 (10.00%)			
Place of residence, *p*-value ^1^		*p* 0.698	*p* 0.669	*p* 0.065
City, *n* (%), M ± SD	11 (7.33%)	0.81 ± 0.68	6.04 ± 1.63	7.06 ± 1.71
Countryside, *n* (%), M ± SD	139 (92.67%)	0.7 ± 0.61	6.09 ± 2.17	8 ± 1.79

^1^ The Mann-Whitney test. ^2^ The Spearman’s correlation coefficient. ^3^ The Kruskal-Wallis test. ^4^ The percentage does not add up to 100 as it was a multiple choice question. M—mean. SD—standard deviation.

**Table 2 ijerph-17-06944-t002:** The work characteristics of respondents compared with the rationing of nursing care.

Characteristics Nurse’s Work	Participations (*n* = 150)	Princa		
		Rationing Care	Quality of Patient Care	Job Satisfaction
Seniority as a nurse, *p*-Value ^1^		*p* 0.175	*p* 0.818	*p* 0.964
0–10, *n* (%)	51 (34%)	r = −0.12	r = 0.019	r = −0.004
11–20, *n* (%)	46 (30.67%)			
21–30, *n* (%)	43 (28.67%)			
31–40, *n* (%)	6 (4%)			
No answer	4 (2.67%)			
Seniority in the intensive care unit and anesthesiology department, *p*-value ^1^		*p* 0.299	*p* 0.803	*p* 0.782
0–5, *n* (%)	50 (33.33%)	r =− 0.091	r =− 0.021	r = 0.782
6–10, *n* (%)	28 (18.67%)			
11–15, *n* (%)	29 (19.33%)			
16–25, *n* (%)	14 (9.33%)			
21–25, *n* (%)	15 (10.00%)			
26–30, *n* (%)	13 (8.67%)			
31–35, *n* (%)	1 (0.67%)			
Number of jobs held, *p*-Value ^2^		*p* 0.253	*p* 0.461	*p* 0.076
One, *n* (%), M ± SD	65 (43.33%)	0.7 ± 0.64	5.88 ± 1.88	7.02 ± 1.62
Two, *n* (%), M ± SD	70 (46.67%)	0.92 ± 0.7	6.11 ± 1.41	7.03 ± 1.83
≥ Three, *n* (%), M ± SD	14 (9.33%)	0.64 ± 0.66	6.43 ± 1.87	8.14 ± 1.46
No answer	1 (0.67%)			
Work system, *p*-Value ^2^		*p* 0.197	*p* 0.606	*p* 0.443
Single shift work, *n* (%), M ± SD	15 (10.00%)	0.74 ± 0.92	5.8 ± 1.9	6.73 ± 1.83
Shift work in 8-hour system, *n* (%), M ± SD	8 (5.33%)	1.13 ± 0.4	6.5 ± 1.07	6.75 ± 1.28
Shift work in 12-hour system, *n* (%), M ± SD	127 (84.67%)	0.79 ± 0.67	6.05 ± 1.68	7.2 ± 1.74

^1^ The Spearman’s correlation coefficient. ^2^ The Kruskal-Wallis test. M—mean. SD—standard deviation.

**Table 3 ijerph-17-06944-t003:** Modified Fatigue Impact Scale (MFIS) compared with the rationing of nursing care.

MFIS Feature	Participations (*n* = 150)	Princa		
		Rationing Care	Quality of Patient Care	Job Satisfaction
Total MFIS result, M ± SD	30.9 ± 15.68			
Physical functions, M ± SD	13.4 ± 7.04			
Cognitive functions, M ± SD	13.99 ± 7.46			
Psycho-social functions, M ± SD	3.5 ± 2.12			
Fatigue, *p*-Value ^1^		r = 0.469 *p* <0.001 *	r = −0.06 *p* 0.469	r = −0.195 *p* 0.017 *
Physical fatigue, *p*-Value ^1^		r = 0.455 *p* <0.001 *	r = −0.041 *p* 0.622	r = −0.195 *p* 0.017 *
Cognitive fatigue, *p*-Value ^1^		r = 0.43 *p* <0.001 *	r = −0.065 *p* 0.432	r = 0.203 *p* 0.013 *
Psycho-social fatigue, *p*-Value ^1^		r = 0.477 *p* <0.001 *	r = −0.039 *p* 0.639	r = −0.105 *p* 0.201

^1^ The Spearman’s correlation coefficient. M—mean. SD—standard deviation. * *p*–Value (<0.05).

**Table 4 ijerph-17-06944-t004:** The regression model and rationing of nursing care.

Feature		Parameter	95%CI		*p*-Value ^1^
Total MFIS result		0.021	0.014	0.028	<0.001 *
Sex					
	Females	Ref.			
	Males	0.082	−0.386	0.549	0.732
Age	[Years]	−0.001	−0.051	0.049	0.966
Marital status					
	Married	Ref.			
	Single	−0.035	−0.3	0.23	0.797
Place of residence					
	City	Ref.			
	Countryside	−0.121	−0.550	0.317	0.588
Education					
	Medium	Ref.			
	Bachelor’s degree	0.085	−0.349	0.519	0.702
	Master’s degree	−0.068	−0.505	0.368	0.759
Qualification courses					
	No	Ref.			
	Yes	−0.166	−0.424	0.092	0.21
Specialist courses					
	No	Ref.			
	Yes	0.041	−0.188	0.27	0.726
Specialization					
	No	Ref.			
	Yes	0.018	−0.237	0.272	0.892
Postgraduate education					
	No	Ref.			
	Yes	−0.135	−0.581	0.311	0.555
Seniority as a nurse					
	[Years]	−0.005	−0.053	0.043	0.851
Seniority in the intensive care unit					
	[Years]	0.002	−0.02	0.025	0.837
Do you have more than one workplace?					
	No	Ref.			
	2 places of work	0.189	−0.046	0.423	0.118
	≥3 places of work	0.13	−0.343	0.604	0.591
Works system					
	Single shift work	Ref.			
	8-hour shift work	0.198	−0.507	0.902	0.583
	12-hour shift work	0.059	−0.405	0.523	0.804

^1^ The multi-factorial linear regression. * statistically significant relationship (*p* < 0.05). CI—confidence interval.

**Table 5 ijerph-17-06944-t005:** The regression model and assessment of the quality of patient care.

Feature		Parameter	95%CI		*p*-Value ^1^
Total MFIS result		−0.013	−0.032	0.006	0.187
Sex					
	Females	Ref.			
	Males	−0.285	−1.566	0.996	0.663
Age	[Years]	−0.065	−0.19	0.06	0.312
Marital status					
	Married	Ref.			
	Single	−0.117	−0.839	0.605	0.752
Place of residence					
	City	Ref.			
	Countryside	−0.088	−1.278	1.103	0.885
Education					
	Medium	Ref.			
	Bachelor’s degree	0.119	−0.987	1.224	0.834
	Master’s degree	0.029	−1.084	1.141	0.959
Qualification courses					
	No	Ref.			
	Yes	−0.2	−0.898	0.498	0.575
Specialist courses					
	No	Ref.			
	Yes	−0.46	−0.683	0.591	0.888
Specialization					
	No	Ref.			
	Yes	0.223	−0.465	0.911	0.527
Postgraduate education					
	No	Ref.			
	Yes	0.868	−0.323	2.059	0.156
Seniority as a nurse					
	[Years]	0.069	−0.053	0.19	0.271
Seniority in the intensive care unit					
	[Years]	−0.02	−0.079	0.039	0.51
Do you have more than one workplace?					
	No	Ref.			
	2 places of work	0.249	−0.384	0.882	0.443
	≥3 places of work	0.044	−1.195	1.284	0.945
Works system					
	Single shift work	Ref.			
	8-hour shift work	1.007	−0.669	2.683	0.241
	12-hour shift work	0.316	−0.796	1.428	0.579

^1^ The multi-factorial linear regression. CI—confidence interval.

**Table 6 ijerph-17-06944-t006:** The regression model and rationing of job satisfaction assessment.

Feature		Parameter	95%CI		*p*-Value ^1^
Total MFIS result		−0.019	−0.038	0	0.057
Sex					
	Females	Ref.			
	Males	0.335	−0.936	1.606	0.606
Age	[Years]	0.021	−0.103	0.145	0.736
Marital status					
	Married	Ref.			
	Single	0.389	−0.328	1.106	0.289
Place of residence					
	City	Ref.			
	Countryside	1.184	0.002	2.365	0.052
Education					
	Medium	Ref.			
	Bachelor’s degree	−0.549	−1.646	0.549	0.329
	Master’s degree	−0.534	−1.638	0.57	0.345
Qualification courses					
	No	Ref.			
	Yes	−0.19	−0.883	0.502	0.591
Specialist courses					
	No	Ref.			
	Yes	−0.098	−0.731	0.534	0.761
Specialization					
	No	Ref.			
	Yes	0.355	−0.328	1.038	0.31
Postgraduate education					
	No	Ref.			
	Yes	−0.058	−1.24	1.125	0.924
Seniority as a nurse					
	[Years]	−0.037	−0.158	0.084	0.549
Seniority in the intensive care unite					
	[Years]	0.019	−0.04	0.078	0.525
Do you have more than one workplace?					
	No	Ref.			
	2 places of work	−0.107	−0.735	0.521	0.74
	≥3 places of work	1.12	−0.109	2.35	0.077
Works system					
	Single shift work	Ref.			
	8-hour shift work	0.433	−1.231	2.096	0.611
	12-hour shift work	0.69	−0.414	1.793	0.223

^1^ The multi-factorial linear regression. CI—confidence interval.

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
