# Peer review of "Rationing of Nursing Care in Intensive Care Units"

_ijerph, 2020, doi:10.3390/ijerph17196944_

Round 1

Reviewer 1 Report

Thank you very much for giving me the opportunity to review your manuscript. My comments are as follows. Hope it helps.

  1. Abstract:

1) Avoid abbreviation in your abstract. Present full-term.

2) Considering the purpose you described and the results you presented, these are not matching. Your purpose was to assess the rationing of nursing care and your results presented the related factors or predictors of rationing of nursing care. Revise the purpose and results consistently in your abstract and in the body of the manuscript.

2. Introduction

1) It is not clear what you wanted to describe in the introduction. I guess that you may want to emphasize to assess rationing of nursing care in nursing practice and what related factors of rationing of nursing care were identified so far. And you need to describe  what problems nurses' fatigue can cause and how fatigue is associated with rationing of nursing care.

Please revise the introduction to show flow above if you like.

2) Line 54. Therefore, rationing of care cause adverse events. Specifically what are the adverse events? What is causal relationship in the context? Add the reference you addressed. You used a couple of times of "therefore" in the manuscript but it is a little confusing to estimate your thoughts in the contexts. Please clarify them in the sentences you addressed.

3. Materials and Methods

1) Add study design.

2) In the instruments, place references of measures in the first sentences. And add reliability and/or validity of the measures.

3) Add data collection procedures.

4) You performed linear regression but why the results were presented in the APPENDIX? I think that you ultimately wanted to say the results of APPENDIX A,B, and C.... Then, you need to present them in the tables not in the Appendix and describe them in the results.

4. Results.

1) Table 1: Why did you present r values in the PRINCA of age groups in stead of mean anSD? Why Kruskal-Wallis test results were not presented? Confusing...

2) Table 2: It is same above. Why did you choose to present r for seniority as a nurse and in the department?

3) Table 3: Why the association between total MFIS and PRINCA was not presented? According to the results of APPendix A, B, and C, you used total MFIS...

5. Discussion

You mentioned your purpose of the study as assessment of rationing of nursing care. I think that you wanted to say the level of rationing of nursing care and the association of fatigue with rationing of nursing care, based on the results you presented. Then, you need to discuss the two things at least: level of rationing of nursing care and how fatigue was associated with rationing of nursing care. And do not just describe your thoughts but present the references based on literature findings to show your thoughts scientifically.

Author Response

Dear Reviewer,

We thank you very much for the new insightful review and valuable suggestions.

Below, we explain point by point the new revised details of the corrections in the manuscript:

  1. Abstract:

1) Avoid abbreviation in your abstract. Present full-term.

Our answer: We improved the abstract. We avoid abbreviations and we present full – term.

2) Considering the purpose you described and the results you presented, these are not matching. Your purpose was to assess the rationing of nursing care and your results presented the related factors or predictors of rationing of nursing care. Revise the purpose and results consistently in your abstract and in the body of the manuscript.

Our answer: We improved the purpose of study.

  1. Introduction

1) It is not clear what you wanted to describe in the introduction. I guess that you may want to emphasize to assess rationing of nursing care in nursing practice and what related factors of rationing of nursing care were identified so far. And you need to describe  what problems nurses' fatigue can cause and how fatigue is associated with rationing of nursing care.

Please revise the introduction to show flow above if you like.

Our answer: In the introduction, we wanted to explain the specifics of a nurse's work in an intensive care unit. The care rationing factors are known to the entire group of nurses, and we wanted to focus on a specific elements of the nursing work in intensive care units. But we consider the reviewer's suggestions very important and valuable, so we developed the issue of fatigue and adverse situations in the introductions. Thank you for this suggestion. We improved the introduction.

2) Line 54. Therefore, rationing of care cause adverse events. Specifically what are the adverse events? What is causal relationship in the context? Add the reference you addressed. You used a couple of times of "therefore" in the manuscript but it is a little confusing to estimate your thoughts in the contexts. Please clarify them in the sentences you addressed.

Our answer: The specifically adverse events are listed in page 2 line 54-56. We also added new and we developed this issue. There was a dot in the middle of this sentence so it changed the context- we improved this.

  1. Materials and Methods

1) Add study design.

Our answer: We added the study design

2) In the instruments, place references of measures in the first sentences. And add reliability and/or validity of the measures.

Our answer: We added in the instruments, the place references of measures in the first sentences and we also added the reliability and/or validity of the measures.

3) Add data collection procedures.

Our answer: We added and the data collection procedures was described in page 3 line 102-106 (in statistic procedure).

4) You performed linear regression but why the results were presented in the APPENDIX? I think that you ultimately wanted to say the results of APPENDIX A,B, and C.... Then, you need to present them in the tables not in the Appendix and describe them in the results.

Our answer: We changed as the Reviewer suggestions was: we changed the appendix as a Table.

  1. Results.

1) Table 1: Why did you present r values in the PRINCA of age groups in stead of mean anSD? Why Kruskal-Wallis test results were not presented? Confusing...

Our answer: Please notice that Table 1 contains the p- value2 (is over the r) and age is counted by the test the Spearman's correlation coefficient (see the footer of the table). Kruskal-Wallis test is used to count the education and it is described as p-value3 (the footer contains the explanation).

And the footer:

The mean and SD of the age is described in main text (page 3 line 120: “The average of age of the study group was  38.75 ± 9.1 years”)

Also in the main text is explain the statistic procedure: “Comparison in three or more groups was made using the Kruskal-Wallis test. Correlations between quantitative variables were analyzed using the Spearman correlation coefficient”.

2) Table 2: It is same above. Why did you choose to present r for seniority as a nurse and in the department?

Our answer: Seniority was a quantitative variables factor so it was using the Spearman correlation coefficient. Please, notice:

And the footer

3) Table 3: Why the association between total MFIS and PRINCA was not presented?

Our answer: It is presented in Table 3.

According to the results of APPendix A, B, and C, you used total MFIS...

Our answer: We used the multifactorial analysis of the independent impact of many variables on the quantitative variable was performed by linear regression. But it is a separate specific analysis, so we fought that put this data in Appendix it will be more transparent.

  1. Discussion

You mentioned your purpose of the study as assessment of rationing of nursing care. I think that you wanted to say the level of rationing of nursing care and the association of fatigue with rationing of nursing care, based on the results you presented. Then, you need to discuss the two things at least: level of rationing of nursing care and how fatigue was associated with rationing of nursing care. And do not just describe your thoughts but present the references based on literature findings to show your thoughts scientifically.

Our answer: We developed the discussion about the rationing of nursing care with the fatigue feature. We also added the literature findings and we developed the references.

Thank you for your important attention.

Dear Reviewer

Once again, thank you very much for the new review and we hope that our new answers are also satisfactory to you.

We really appreciate it. Thanks to these changes, our article gained new value.

Best regards,

Authors

Reviewer 2 Report

I believe that the problem of the article is relevant and current in nursing staff in ICU.

There is detailed description of the research methods used. The design is correct and it is possible to validate the veracity of the results

The sample is insufficient as you have explained in the limitations

The results shown are concrete and detailed, explaining how to obtain this information and what scientific evidence it has.

The discussion is extensive and reasoned

Good job

Author Response

Our answer:

Dear Reviewer

We thank you very much for the review - we really appreciate it. Thank you for yours important attention and valuable opinion.

Best regards,

Authors

Reviewer 3 Report

This is an interesting paper describing the impact of nursing shortage on quality of care provided at intensive care units in Poland. The ‘s, that the nursing fatigue is the most important factor could be well predicted. It would be more important to provide knowledge about nursing activities that are actually affected by nursing shortage. Was there any evidence of patient’s harm caused by this rationing?

The manuscript need careful editing of English language (some of the examples are below). Also, past tense should be always used when reporting about study results.

Page 2, lines 79-80: …. author's survey questionnaire consisting of 10 questions constituting the metric, 13 open and closed questions… How many questions were there? 10 or 13?

Page 2, lines 89-90: unclear sentence  “In these two questions about the nursing assessment 90 of the quality of patient care and job satisfaction.”

Page 3, line 97: instead of …. many points we are talking about high fatigue… write ….. many points indicate high fatigue ….

Page 3, lines 98-100: unclear

Page 3, line 120: … average AGE of the study group …

Page 5, line 156: … The remaining respondents are one-shift employees ….. What is that?

Page 6, line 181: range between "rare"…..  and what?

Author Response

Dear Reviewer,

We thank you very much for the new insightful review and valuable suggestions.

Below, we explain point by point the new revised details of the corrections in the manuscript:

Comments and Suggestions for Authors

This is an interesting paper describing the impact of nursing shortage on quality of care provided at intensive care units in Poland. The ‘s, that the nursing fatigue is the most important factor could be well predicted. It would be more important to provide knowledge about nursing activities that are actually affected by nursing shortage. Was there any evidence of patient’s harm caused by this rationing?

Our answer:

 The study of evidence of harm to patients from unfinished nursing care was not a goal of this study. The purpose of this research was to understand the impact of sociodemographic factors, the impact of nursing staff fatigue, and employment factors on rationing care in the intensive care unit. Fatigue as a factor in rationing care is described in the literature, however the innovation of this work is that we have studied it among a small specific group of anesthesia nurses. The specificity of the work of an inactive care nurse differs from that of a nurse in other departments, and thus factors such as fatigue and burnout are also subject to this dependence.

The manuscript need careful editing of English language (some of the examples are below). Also, past tense should be always used when reporting about study results.

Our answer:

Thank you for Your valuable comments, of course we will fixed it as expected by the Reviewer

Page 2, lines 79-80: …. author's survey questionnaire consisting of 10 questions constituting the metric, 13 open and closed questions… How many questions were there? 10 or 13?

Our answer:

Thank You for the suggestion, we have corrected this sentence as:

“author's survey questionnaire consisting of 13 questions constituting the metric, 3 open and 10 closed questions…”

Page 2, lines 89-90: unclear sentence  “In these two questions about the nursing assessment 90 of the quality of patient care and job satisfaction.”

Our answer:

This sentence should be without the dot- this is one sentence.

“ In these two questions about the nursing assessment of the quality of patient care and job satisfaction the scale of answers is from 0 to 10, where higher numbers mean better nursing assessment of the quality of patient care and greater job satisfaction”

Page 3, line 97: instead of …. many points we are talking about high fatigue… write ….. many points indicate high fatigue ….

Our answer:

Thank You for the suggestion, we have corrected this sentence.

Page 3, lines 98-100: unclear

Our answer:

Thank You for the suggestion, sentence is translated incorrectly. We improved that.

Page 3, line 120: … average AGE of the study group

Our answer:

Thank You for the suggestion, we have corrected this sentence, as:

“The average age of the study group was”

Page 5, line 166: … The remaining respondents are one-shift employees ….. What is that?

Our answer:

One-shift employees – that means - employees that working in one shift

Page 6, line 181: range between "rare"…..  and what?

Our answer:

We andede “ sometimes”

It missed a word. Thank You very much for Yours Comments.

Submission Date

25 August 2020

Date of this review

01 Sep 2020 22:37:36

Thank you for your important attention.

Dear Reviewer

Once again, thank you very much for the new review and we hope that our new answers are also satisfactory to you.

We really appreciate it. Thanks to these changes, our article gained new value.

Best regards,

Authors

Round 2

Reviewer 1 Report

Overall, it has been much improved but consider the following:

  1. Reliability: present exact values in the developer's study and your study results.
  2. Discussion: it has been some improved but my comments were not considered. In the discussion section, you need to discuss your results in depth... It should be the main stream but most of your discussion is description from others' study results. It is not discussion. For example, in lines 247-249 of page 10, you presented the results lastly in a contaxt and you did not discuss why the differences betwee your study results and others were not described. Instead, you need to describe your results first in line 242-244 and then, you can describe the other studies' findings. And then discuss why the differences were showed. And for example, if this difference is of clinical and practical significance, describe what it is.

Author Response

Dear Reviewer,

We thank you very much for the new insightful review and valuable suggestions. We read it and working very carefully to understand Yours comments. Below you find the improved manuscript. We hope that our new improved section of discussion are now satisfactory to you.

Best regards,

Authors
